# Metabolomics-Driven Biomarker Discovery for Breast Cancer Prognosis and Diagnosis

**DOI:** 10.3390/cells14010005

**Published:** 2024-12-25

**Authors:** Rasanpreet Kaur, Saurabh Gupta, Sunanda Kulshrestha, Vishal Khandelwal, Swadha Pandey, Anil Kumar, Gaurav Sharma, Umesh Kumar, Deepak Parashar, Kaushik Das

**Affiliations:** 1Department of Biotechnology, Institute of Applied Sciences & Humanities, GLA University, Chaumuhan, Mathura 281406, Uttar Pradesh, India; rasanpreet864@gmail.com (R.K.); sunanda.kulshrestha@gmail.com (S.K.); vishal.khandelwal@gla.ac.in (V.K.); swadhap547@gmail.com (S.P.); 2Division of Hematology & Oncology, Department of Medicine, Medical College of Wisconsin, Milwaukee, WI 53226, USA; dparashar@mcw.edu; 3National Institute of Immunology, New Delhi 110067, India; anilk@nii.ac.in; 4Cardiovascular and Thoracic Surgery, University of Texas Southwestern Medical Center, Dallas, TX 75390, USA; gaurav.sharma@utsouthwestern.edu; 5Advanced Imaging Research Center (AIRC), University of Texas Southwestern Medical Center, Dallas, TX 75390, USA; 6Biomedical Engineering, University of Texas Southwestern Medical Center, Dallas, TX 75390, USA; 7Department of Biosciences, Institute of Management Studies Ghaziabad (University Courses Campus), Ghaziabad 201015, Uttar Pradesh, India; umeshkumar82@gmail.com; 8Biotechnology Research and Innovation Council-National Institute of Biomedical Genomics, Kalyani 741251, West Bengal, India

**Keywords:** metabolomics, breast cancer, therapeutics, biomarkers, metastasis

## Abstract

Breast cancer is a cancer with global prevalence and a surge in the number of cases with each passing year. With the advancement in science and technology, significant progress has been achieved in the prevention and treatment of breast cancer to make ends meet. The scientific intradisciplinary subject of “metabolomics” examines every metabolite found in a cell, tissue, system, or organism from different sources of samples. In the case of breast cancer, little is known about the regulatory pathways that could be resolved through metabolic reprogramming. Evidence related to the significant changes taking place during the onset and prognosis of breast cancer can be obtained using metabolomics. Innovative metabolomics approaches identify metabolites that lead to the discovery of biomarkers for breast cancer therapy, diagnosis, and early detection. The use of diverse analytical methods and instruments for metabolomics includes Magnetic Resonance Spectroscopy, LC/MS, UPLC/MS, etc., which, along with their high-throughput analysis, give insights into the metabolites and the molecular pathways involved. For instance, metabolome research has led to the discovery of the glutamate-to-glutamate ratio and aerobic glycolysis as biomarkers in breast cancer. The present review comprehends the updates in metabolomic research and its processes that contribute to breast cancer prognosis and metastasis. The metabolome holds a future, and this review is an attempt to amalgamate the present relevant literature that might yield crucial insights for creating innovative therapeutic strategies aimed at addressing metastatic breast cancer.

## 1. Introduction

Breast cancer is the most frequent malignancy in women and one of the main causes of cancer-related deaths. There is significant evidence that various risk factors increase women’s susceptibility to breast cancer. Nonetheless, the prevalence of breast cancer among women worldwide is growing [1]. Late menopause, early menarche, and age at first full-term pregnancy are all reproductive variables that increase breast cancer risk [2]. Aside from these, other risk factors for breast cancer include obesity and a first-degree family history [3]. A family history of breast cancer raises the risk of developing breast cancer by 2.5 times or more [4]. According to K.B. Kuchenbaecker et al., women are also considerably more at risk if they have both BRCA1 and BRCA2 mutations [5]. A high birth weight, alcohol use, and body fat are a few other recognised risk factors [2]. Obese postmenopausal women have a 40% higher risk of breast cancer [6]. Even non-genetic, changeable risk variables, such as insurance status, education level, and income, can influence a person’s chance of developing breast cancer [7]. Another important risk factor is ethnicity; non-Hispanic white females are more likely to develop breast cancer than non-Hispanic black females [8]. It is interesting to note that most women receiving a new breast cancer diagnosis do not have any established risk factors [9].

According to predictions from the World Health Organization, 12% of newly diagnosed malignancies globally in 2021 were breast cancers. Globally, there will likely be over 3.2 million new instances of breast cancer a year by 2030, as per the study by O. Ginsburg et al. [10]. Based on the stage and subtype of the cancer, there are differences in the five-year survival rate [11].

## 2. Metabolome and Metabolomics

The total amount of metabolites—which serve a variety of purposes—found in a cell, tissue, organ, or organism is known as the metabolome. The study of the metabolites in the metabolome is the focus of the comparatively recent field of metabolomics [12]. The end products of both anabolism and catabolism are included in the metabolome. Like the fields of proteomics, transcriptomics, and genomes, metabolomics research has grown in significance.

An organism’s phenotype is intimately associated with its metabolites; an organism’s DNA primarily defines its metabolome. Measuring the magnitude of low-molecular-weight metabolites in biological systems, which represents the vigorous response to genetic alteration, is known as metabolic profiling or metabolomics. The information contained in an organism’s DNA (genome) is translated into RNA (transcriptome), then into proteins (proteome), and ultimately into tiny molecules known as metabolites (metabolome), which are the product of metabolism. Therefore, any alteration to a gene, whether it is mutation, over-expression, or under-expression, will impact an organism’s metabolomics profile. Gene expression profile changes are the cause of many diseases, including cancer. For example, the BRAC1/BRAC2 genes are the most frequently altered in hereditary breast cancer [13]. Thus, genetic mutations may result in modifications to the metabolic profile, which may ultimately promote the growth of cancer.

Metabolomics is regarded as an extremely potent, dependable instrument with great repeatability that may greatly influence people’s health [14]. A thorough metabolomics investigation of 928 cell lines from over 20 distinct cancer types identified 225 compounds related to cancer metabolism [15]. Metabolite level changes can serve as prognosis indicators [16], therapeutic targets [17,18], and diagnostic markers [19,20]. A person’s metabolomics profile might change due to a variety of biological processes related to gender, age, obesity, medicine, cancer, diabetes, cardiovascular disease, etc. [21]. The metabolome profile can be utilised as a marker to envisage a newborn’s future probability of cardiovascular disease if their mother has gestational diabetes, according to a study by Mansell et al. [22]. Furthermore, it has been shown by epidemiological and experimental evidence that geriatric syndromes, age-related disorders, and aging itself are associated with significant metabolite alterations [23]. It is commonly recognised that several cancer forms are associated with old age. Amino acid metabolism, redox homeostasis, lipid metabolism, and nutrition sensing have all been linked to endocrine and mitochondrial processes, calorie constraints, and signalling pathways in aging [24,25]. Studying the metabolome will be crucial to the advancement of personalised medicine, as well as the early identification and diagnosis of diseases, as alterations in metabolomics profiles can be related to a variety of pathological disorders.

### Analysis of the Metabolome

The following three main methods are available for measuring metabolites: Nuclear Magnetic Resonance Spectroscopy (NMR), Liquid Chromatography (LC), and Gas Chromatography–Mass Spectrometry (GC-MS) [26]. According to U.L. Günther, 2015, NMR is a dependable and effective method that can identify even minute variations in the concentrations of metabolites in biological materials. NMR is often considered to be less sensitive compared to Mass Spectrometry (MS) [27]. As a plus point, NMR allows for the reuse of samples, which can be a significant advantage in longitudinal studies and cases where sample quantity is limited.

On the other hand, in mass spectrometry, there are the following two methods: targeted and untargeted. An untargeted strategy concentrates on a large variety of metabolites, which are subsequently recognised and described from the samples. The goal of a focused strategy is to identify the pre-identified and described metabolites based on their mass. The focused approach is often more sensitive and selective. However, there is always a challenge associated with these techniques, similarly in the case of GC/MS, which taps only into the volatile compounds that are present in samples and compromises metabolites that are non-volatile but might be significant, while in LC/MS and the associated addition of sophisticated MS, there are greater chances of acquiring errors and the identification of noise (unwanted particles), along with only the one time use of samples [28].

Scanning Electron Microscopy (SEM), Matrix-Assisted Laser Desorption Ionization (MALDI), and Nanostructure Imaging Mass Spectrometry (NIMS) can be used to execute and visually view metabolomics profiling [29,30]. DeviumWEb, XCMS Online, and MetaboAnalyst are just a few of the platforms used to handle the obtained data in preparation for statistical analysis [29]. Following their identification, metabolites are linked to a certain phenotype, physiological condition, or anomaly [29]. Another method for identifying metabolites is to compare the mass spectral results with databases like METLIN, NIST05, Mass Bank, GOLM, or the most popular one, the Human Metabolome Database (HMDB) [29,31]. Even while metabolomics seems very promising, there are a few drawbacks. A total of 247 inborn error metabolites (IEMs) for humans, comprising both exogenous and endogenous metabolites, were included in the latest issue by the HMDB [32]. The extraction, purification, fractionation, and identification of metabolites pose additional difficulties. The last option is the detection of false-positive metabolites [33]. Though it is still in its infancy, the area of metabolomics has great potential for discovering novel pathologies, biomarkers, and treatment targets. Some of the reported work along with the detection techniques are given in Table 1.

## 3. Unravelling Breast Cancer in the Era of Multi-Omics

The core tenets of molecular biology state that the genetic information encoded in DNA (genome) is converted into proteins (proteome), metabolites (metabolome), and RNA (transcriptome) [35,36] Reversible modifications occur in both histone proteins (epigenome) and DNA, and a similar process occurs in RNA (epi transcriptome) and proteins (epi proteome) [37]. As a result, different kinds of molecular data—like the novel ideas put forward thus far—are becoming available for the same collection of clinical samples [38]. At the monogenic level, inherited mutations in BRCA1/2, the susceptible tumour suppressor genes for breast cancer (BC), among other cancers [39], have long been contemplated as the main genetic constituents in BC [40]. This concept dates to a time when a gene function was considered in isolation from its networking partners. New, cutting-edge genomics-based techniques engaged in the analysis of entire genomes have replaced the outdated genetics-based processes that examined individual variations or single genes due to the advancements in high-throughput technology [41]. Table 2 gives a summary of the omics-based investigations of BC samples using different technologies, along with their concluding remarks.

With the other molecular levels involved, including the transcriptome, proteome, translatome, and interactome, the investigation of the genetic underpinnings of BC cancer become more intricate in this setting [52]. Thus, genomics has shown molecular changes in single genes, gene panels, or entire genomes, and genomics-guided biomarker testing has found mutations in many BC candidate genes [53]. Next, studies based on transcriptomics and genomics have characterised novel genome-driven integrated categorisations of BC, defining integrative clusters linked to specific molecular drivers, oncogenic pathways, and distinct clinical outcomes [54]. Furthermore, as noted by D. Hamdan et al., genomics plays a direct role in analysing the heterogeneity of BC tumours and is also utilised in managing triple-negative BC and HER2 overexpression [55].

According to Y. Hasin et al., high-throughput sequencing technologies have transformed the domains of transcriptomics and genomics, hence advancing the area of biomedical research that employs multi-omics approaches to illness [41]. Parallel DNA and RNA sequencing techniques have produced large-scale data on hundreds of BC genomes [56]. However, across a range of Mendelian disorders, the genomics-based diagnosis rates are only about 50% [57]. Genome-wide association studies (GWASs) have been used to analyse point mutations with a high penetrance. GWASs search through hundreds to thousands of genetic variants across genomes to identify those that are considerably linked to a particular disease and to identify the genetic risk factors that are common in a target population. More than 200 susceptibility loci for BC have been found by GWASs [58]. Limited relevance for gene–environment interfaces in BC risk was reported by research based on transcriptome-informed genome-wide gene–environment interactions [59]. However, like in the cases of gene–lifestyle interaction and pro-inflammatory signalling, these investigations help to comprehend the interplay between environmental risk factors and genetic variations in BC [60]. The characterisation of the various signalling pathways connected to BC development in women of European and Asian heritage [61] and the identification of unique loci linked to mammographic density phenotypes [62] have also been accomplished using GWASs and transcriptome-wide association studies (TWASs). The goal of gene silencing/RNA interference (RNAi)-pathway-based studies is to correlate the various BC types and stages with RNAi categories, such as microRNA (miRNA) and small interfering RNAs (siRNAs), relative to healthy cells, highlighting the high potential of these RNAi categories for monitoring, diagnosis, and treatment [63]. Therefore, a major factor in the understanding of gene function is the post-translational regulatory mechanism, which is controlled by siRNAs and miRNAs and inhibits gene expression in cancer and other disorders [64,65].

The proteome expression of the proteome more closely represents the crucial alterations in the pathophysiology of a tumour, even if the tumour transcriptome and genome are valuable resources for the identification of new biomarkers for BC. Personalised medicine can be advanced by more extensive insights into alterations in the proteome because of high-throughput Mass Spectrometry (MS)-based approaches [66]. Furthermore, proteomics and RNA-seq complement each other well in identifying the functional significance of uncommon genetic variants, as shown by GWASs [57]. Over the past several decades, a variety of omics-based techniques have made major progress in the quest for non-invasive biomarkers for all-stage, and particularly early-stage, BC diagnosis in cancer liquid biopsies, therefore avoiding invasive tumour tissue biopsies or operations. Blood/plasma-based genomics, which typically entail circulating tumour DNA (ctDNA) or cell-free DNA (cfDNA) analyses, are helpful for a variety of purposes, including pre-diagnosis [67], dormancy [68], sub-clonal variation in advanced BC [69], the prediction of disease-free survival (DFS) [70], and assessments of TNBC progression and personalised management and diagnoses in patients.

## 4. Biofluids as Detection for Biomarkers in Breast Cancer

Blood-based DNA methylation indicators may be a useful tool for BC risk stratification, according to studies on blood-based epigenomics [71,72]. Numerous transcriptomics studies based on plasma have been extensively involved in the process of finding biomarkers for BC diagnoses. These studies use qRT-PCRs, circulating micro-RNAs qRT-PCRs, and plasma-derived exosomal circular RNAs-sequencing (circRNA-seq), as well as exosomal long RNA-sequencing (exLR-seq) [73,74]. To distinguish BC from other disorders, phosphopeptides and plasma peptides can be identified using LC-ESI-MS/MS-based approaches [75]. New diagnostic biomarkers for BC can be found by metabolomics-based methods [76]. Figure 1 summarises the biomarkers which could be studied for the early detection of BC and are non-invasive in nature.

Liquid Chromatography–Mass Spectrometry (LC-MS) and LC-MS/MS are used in targeted plasma-based metabolomics to identify metabolic biomarkers for early-stage breast cancer detection. The plasma metabolite profiles of BC patients were compared to healthy controls, highlighting the potentially dysregulated pathways contributing to BC pathogenicity [77]. Glutamate, sphingomyelins, and cysteine have been highlighted as possible diagnostic biomarkers for BC, as evidenced by the unique alterations in the proteomic and metabolic profiles of BC patients found by integrative studies of plasma metabolomics and proteomics [78]. Tumour-subtype-specific biomarkers in BC interstitial fluid can be found using LC-MS/MS-based proteomics [79]. Studies utilising lipidomics have examined the relationship between the lipid BC characteristics found in tissue by Desorption Electrospray Ionization–Mass Spectrometry Imaging (DESI-MSI) and those found in BC patients’ plasma by LC-MS [80,81].

Genomic-, epigenomics-, proteomics-, and metabolomics-based techniques can identify cellular alterations in the early stages of breast cancer through non-invasive sampling. This helps to develop non-invasive diagnostic tests. An epigenomics-based analysis of nipple aspirate fluid (NAF) from healthy women revealed the hypermethylation of genes absent in both NAF and benign breast tissue in patients with DCIS stage I [82,83] (Figure 1). Thus, it has been suggested that promoter hypermethylation is a possible biomarker for BC. Additionally, multi-omics assessments of BC patients may benefit from the identification, extraction, and characterisation of components produced from breast tumours in saliva [84].

Saliva is a promising, non-invasive source of protein biomarkers and dysregulated proteins, which can accurately differentiate BC patients from healthy controls [85] (Figure 1). Saliva has been suggested as a viable substitute for Next-Generation Sequencing (NGS) in the detection of hereditary BC mutations, as it retains high-quality genomic DNA [86,87]. Unique peptide-based signals in saliva have been identified by targeted proteomics-based analysis as potential predictors to differentiate between TNBC and healthy people [88]. The salivary metabolite profiles of women with BC have been revealed by untargeted metabolomics-based techniques and bioinformatics, revealing the up- or down-regulated metabolites linked to BC and potential helpful biomarkers for BC [89]. Bentata et al.’s 2020 study demonstrated the enrichment of genes with functional annotation in alternative splicing, which can indicate BC, uncover the mRNA transcriptome, and create a saliva test [90]. Integrative salivaomics, which encompasses salivary genomics [87], epigenomics, transcriptomics [90], proteomics [88], metabolomics [79], immunomics, and microbiomics approaches, is, therefore, focused on the identification of non-invasive biomarkers [91]. Additionally, based on the molecular biological subtype of BC and the expression levels of HER2, the progesterone receptor (PR), and the oestrogen receptor (ER), salivaomics is helpful for determining the metabolic properties of saliva [92]. Additionally, non-invasive, low-cost, sensitive, and simple to apply in clinical settings, urine-based diagnostic approaches indicate pathological changes, particularly in the early stages of the disease [73,93]. The goal of urinomics is the development of non-invasive biomarkers using methods such as transcriptomics, exosomics, proteomics, and metabolomics [76,94]. According to M. Hirschfeld et al., urinary exosomal miRNAs have been suggested as possible non-invasive indicators in the identification of BC [95]. Breast milk analysis is useful for diagnosing BC, as well as for risk assessment and early diagnosis [96]. Aslebagh et al., 2018, states that milkomics is a method that combines the findings of proteomics-based research to provide potential biomarkers for the early identification of breast cancer [97]. Deregulated circular RNAs (circRNAs) from liquid biopsies, such as milk, may also be clinically significant as prognostic, predictive, and diagnostic biomarkers that are important in the development of breast cancer and breast carcinogenesis [98].

Every omics field has its analytical tools to help provide more thorough data. The gene names that encode the detected proteins, for instance, may be found in the proteomics-based domain. Proteomics experiments utilise technologies like GSEA and STRING for gene set enrichment analysis and search tools for interacting genes/proteins, enabling more comprehensive data processing in various omics fields. The following is a list of some of the online resources and platforms: BigOmics Analytics offers a user-friendly omics toolkit called “Omics Analysis for Everyone”. BioCyc offers large-scale dataset analysis tools for metabolomics, gene expression, proteomics, and other high-throughput data. NetGestalt is an online application for multi-omics data visualisation and integration. MiBiOmics is an interactive web-based application for dynamic association exploration across datasets. Subio Platform is an expert software for quantitative omics data analysis, including transcriptomics, epigenetics, and proteomics.

As a result, there are several ways to analyse multiomics datasets, and many of them can yield more thorough results. Before selecting an omics-based approach, we must consider the desired result. Do we want to find the genome, metabolome, proteome, or other “omes”? There are variances in the duration of sample preparation for the analysis of each omics approach, along with potential drawbacks. Each kind of genomic sequencing—DNA-seq, RT-qPCR, and DNA microarrays—has advantages and disadvantages of its own. Microarrays may identify known genes and transcripts only, but they can identify many samples at a reasonable cost.

Proteomics-based techniques can be time-consuming because of the instrument time required, but they can detect all the expressed proteins in the proteome and pinpoint which proteins are dysregulated under circumstances. Because of their specificity, single-omics and multi-omics techniques are substantially more costly than others.

Apart from this, some of the non-traditional techniques also include biofluid sources such as tears and nipple aspirates. Anna et al., 2022, in their work, gave insights into tear samples for use in breast cancer detection. Breast cancer can affect the quantitative levels of the ocular proteins S100A8, S100A9, and galectin-3-binding protein [99]. Similarly, another paper studied exosome detection in tear samples from breast cancer patients. The study suggested that the tear exosomes contained a much higher quantity of exosome markers (CD9 and CD63) as compared to the serum exosomes. The status of miRNAs, which have been previously reported in serum from patients with metastatic breast cancer, was investigated in tear exosomes using quantitative reverse-transcription polymerase reaction (qRT-PCR) and Western blot analysis. In comparison to healthy volunteers, the tear exosomes from patients with metastatic breast cancer showed elevated expressions of miR-21 and miR-200c, two genes specific to breast cancer, according to qRT-PCR and Western blot analysis [100].

Nipple fluid is also advocated to be a reservoir of metabolomic entities which could be helpful for breast cancer diagnosis. Clinical screening for the early detection of this disease has not been successful in translating the tissue and serum biomarkers that have emerged for the stratification, diagnosis, and predictive outcomes of breast cancer. For biomarker profiling, non-invasive methods exist for collecting breast-specific liquid biopsies, such as nipple aspirate fluid (NAF), a naturally occurring secretion of breast epithelial cells [101].

To provide a thorough knowledge of the molecular alterations that may have an impact on the disease state, cellular response, and development, multi-omics integrates many omics-based methodologies. Since single-cell omics techniques are not practical for large-scale samples, they cannot be used to examine various disease states. Instead, they are limited to reflecting only one feature of a biological system at a single-cell resolution. Because single-cell omics and multi-omics techniques are expensive, it is best to use modest sample numbers. Every omics-based technique has had its advantages and disadvantages highlighted, in addition to the ideal sample size, sensitivity, and cost. As each omics approach provides a distinctive story about the state of a patient’s body, it is impossible to compare them all and decide which is the best.

## 5. Metabolomics-Based Molecular Signatures of Breast Cancer

### 5.1. The Metabolic Signatures of Fundamental Breast Cancer Subtypes

Breast cancer can be classified into the four subtypes of Luminal A, Luminal B, HER-2-enriched, and triple-negative breast cancer [102] based on ki-67 expression and hormone receptor status. Metabolomics can identify these subtypes using eight metabolites—carnitine, proline, alanine, lysophosphatidylcholine (16:1), glycochenodeoxycholic acid, valine, and 2-octenedioic acid, distinguishing between ER-positive or ER-negative individuals and HER2-negative or HER2-positive individuals. This helps in understanding the molecular makeup of breast cancer. HER2-positive individuals had higher amounts of proline, valine, carnitine, lysophosphatidylcholine (20:4), and 2-octenedioic acid than HER2-negative patients did. When comparing the levels of glycochenodeoxycholic acid, alanine, lysophosphatidylcholine (16:1), valine, and 2-octenedioic acid, ER-positive patients showed higher levels than ER-negative patients [103]. Additionally, ER-positive subtypes demonstrated higher levels of glutamine biosynthesis and secretion than ER-negative subtypes, and ER-negative subtypes demonstrated noticeably higher levels of serine metabolism and glutamine utilisation than ER-positive subtypes [104]. Furthermore, ER-negative BC has an increased beta-alanine buildup, whereas the ER-positive subtype exhibits a lower 4-aminobutyrate aminotransferase expression [105]. Furthermore, another investigation noticed that the basal-like and HER-2 subgroups showed clear changes in glutamine and glucose metabolism, and the luminal B subgroup preferred to use fatty acids as an energy source [106]. In thus paper, the epithelial–mesenchymal transition (EMT) was a crucial characteristic indicating the more aggressive behaviour of cancer cells. By employing MALDI-TOF/TOF Mass Spectrometry and NMR to compare the most aggressive breast cancer cell line MDA-MB-231 to the less aggressive breast cancer cell line MCF-7, the lipidomic profiles of both cells were discovered to vary. Compared to MCF7 cells, MDA-MB-231 had a reduced de novo lipid production and an increased triacylglycerol (TAG) metabolism, which may serve as distinguishing factors between luminal and basal-like breast cancer [107,108].

### 5.2. Metabolomics-Based Reclassification of Breast Cancer

Due to the heterogeneity of the malignancy, the clinical response to treatment in BC might differ, even within the same subtype; therefore, a more precise categorisation is required [106]. To monitor the glucose, nucleotide, and amino acid metabolic pathways in the BT474 and MCF-7 cell lines, as well as the MDA-MB-468 and MDA-MB-231 cell lines, investigators added stable isotopic 13C substrates. Significant variations within the same subtype and similarities across subtypes were demonstrated by the metabolites. In [108], hormone receptor-specific energy usage patterns were found in cell lines, while [109] used gene expression microarrays and HR MAS MRS (high-resolution magic angle spinning magnetic resonance spectroscopy) for tissue analysis. These researchers discovered that luminal A may be divided into A1–A3 subgroups. Significant variations were also seen between these three subtypes in the signals originating from glucose, a-H amino acid, alanine, lipid residues, and myo-inositol. Furthermore, a GO keywords enrichment study of these subgroups revealed significant variations in biological processes and molecular activities [109]. The metabolomics-classified BC molecular subtypes can aid in the development of tailored treatment plans. A different study used the Luminal A subtype and divided it into three unique metabolic clusters (1–3) based on the metabolite patterns that differed, mostly in the areas of glutaminolysis, glycolytic activity, and phospholipid metabolism. The prognosis of BC and other tumour biological behaviours were linked to the varying quantities of metabolites in these sub-clusters [49,110].

## 6. Overview of Breast Cancer’s Metabolic Programming

Breast cancer prognosis and metastasis involve a complex network of affected metabolisms, as shown in Figure 2. The process of breast cancer metastasis entails several stages, beginning with the invasion of neighbouring tissues, then intravasation into the lymphatic or blood vessels, movement throughout the body, and extravasation into distant tissues. Tumour metastasis can be caused by several variables, including aberrant TGFb production, MET amplification, and EMT. There is also the involvement of several metabolisms, which are represented in schematic diagram in Figure 3 and given in detail below.

### 6.1. Glucose Metabolism

Normal cells rapidly proliferate, activating various signalling pathways to respond to external growth cues, suppressing oxidative phosphorylation, and promoting glycolysis and anabolic metabolism for cell development. Even in the absence of outside signals, cancer cells can use this process to fulfil their demands for development [111]. Cancer cells coexist with glycolysis and OXPHOS (Oxidative Phosphorylation) to varying degrees, unlike normal cells, where they are always negatively connected. Cancer cells primarily produce ATP through OXPHOS, while normal cells rely on glycolysis for energy production [112,113]. Tumour cells were revealed to have dual metabolic natures, with the ability to transition from aerobic glycolysis to the OXPHOS phenotype in response to lactic acidosis [114]. Additionally, certain tumours display two compartment tumour metabolisms, also known as metabolic coupling or the reverse Warburg effect, which suggests that the nearby cancer cells are sustained by glycolytic metabolism in the cancer-related stroma [115]. In addition to contributing to chemotherapy resistance, this metabolic phenotype explains why certain tumour cells exhibit the paradoxical combination of high mitochondrial respiration and a low glycolysis rate [116]. Research using a large sample size found that luminal subtype correlates with metabolically inactive reverse Warburg/null phenotypes, while TNBC correlates with metabolically active Warburg/mixed phenotypes [117]. Furthermore, induced hypoxia-inducible factor 1 (HIF-1) can enhance glucose metabolism to maintain redox homeostasis, while also increasing the formation of reactive oxygen species (ROS) in breast cancers [118,119].

The glucose transporter proteins (GLUTs) allow glucose to traverse cell membranes, and the expression of distinct GLUTs in breast tumours is associated with varying clinical stages and prognoses. Breast cancer cells have functioning GLUT1-5 and GLUT12 [120], with GLUT1 appearing to have the most significant impact [121]. In [117], it was found that TNBC had the greatest expression of GLUT1 in comparison to other subtypes, indicating a highly active metabolic condition. Furthermore, hexokinase (HK) and lactate dehydrogenase-A (LDHA), two essential glycolysis-related enzymes, are strongly active in breast cancer and linked to the development and spread of the disease [122].

The pentose phosphate pathway (PPP) is a method of oxidative glucose breakdown that produces NADPH, ribose phosphate, and F6P, which cancer cells use to meet their anabolic needs and react to oxidative stress [123]. Various molecular subtypes of breast cancer express PPP-related proteins in various ways. For instance, higher expressions of 6-phosphogluconolactonase (6PGL) and glucose-6-phosphate dehydrogenase (G6PD) suggest that the HER2 subtype of breast cancer has a more active PPP than other subtypes [124]. According to A. Benito et al., there exists a favourable correlation between the expressions of G6PD and transketolase (TKT) and reduced overall and relapse-free survival in breast cancer [125].

### 6.2. Amino Acid Metabolism

In addition to aiding in the provision of energy, glutamine and its metabolic intermediates, such as the antioxidants glutathione (GSH) and nicotinamide adenine dinucleotide (NADH), also assist cells in withstanding oxidative stress, which supports the growth and spread of tumour cells [126]. Glutamine is imported into cancer cells by several transporters, including the Na+-dependent transporters, system ASC, which preferentially transports alanine/serine/cysteine, and the Na+-coupled neutral amino acid transporters (SNATs) [127].

Certain cancer cells have a “glutamine addiction”, indicating that they require exogenous glutamine to live [128]. Oncogenic transcription factors like c-MYC and RAS can upregulate glutamine transporters and enzymes, such as alanine-serine cysteine transporter 2 (ASCT2) and glutaminase (GLS)-1, in cancer cells, increasing their metabolic activity. c-MYC stimulates ASCT2 and GLS-1 expression in response to lactic acid production, thereby enhancing glutamine uptake and catabolism [129,130]. Notably, a metabolomic investigation revealed that the glutamate-to-glutamine ratio (GGR) in breast tumour tissues was greater than that in normal tissues, particularly in tumours that did not express ER. Furthermore, there was a strong correlation between the GGR levels and the tumour grade and ER status [131]. HER2-positive breast cancer was shown to have higher expression levels of glutamine-metabolism-related proteins than other subtypes, including glutamate dehydrogenase (GDH), ASCT2, and GLS-1. This suggests that HER2-positive breast cancer has the greatest levels of glutamine metabolism activity [132].

Other examples of gluatamine addiction by cancer cells have been reported. Glutaminase I (GLS-I) is an enzyme that uses glutaminolysis to convert glutamine to glutamate, which glutamate dehydrogenase then uses to convert to alpha-ketoglutarate. The TCA cycle is initiated by this alpha ketoglutarate. Via the TCA cycle, the glutaminolysis pathway not only produces large amounts of energy, but also the macromolecules needed for the growth and multiplication of cancer cells [111]. Another study gave the hypothesis that, under hypoxic conditions, the glutamine metabolism in tumour cells can change from an oxidative to a reductive carboxylation pathway by entering the TCA cycle. Through reductive carboxylation, tumour cells primarily rely on glutamine-derived α-KG to produce acetyl coenzyme A, citrate, and lipids. This process could be triggered by the HIF-1α transactivating the pyruvate dehydrogenase kinase 1 (PDK1) gene, which would decrease the quantity of pyruvate that enters the TCA cycle [133]. According to A.S. Tibbetts et al., one-carbon metabolism, also referred to as the network of folate utilisation processes, is involved in several metabolic pathways, including methylation and reductive metabolism, de novo nucleotide biosynthesis, and amino acid biosynthesis and degradation [134]. As per [135], there is a consensus that the high rate of tumour cell proliferation is mostly supported by one-carbon metabolism. One-carbon metabolism is crucial for DNA production and methylation, primarily facilitated by folate (vitamin B9) and other B vitamins like B6 and B12 [136]. While there is ongoing debate regarding the association between folic acid intake and breast cancer risk, a recent meta-analysis examining 23 prospective studies discovered that increasing one’s intake of folate reduced the risk of ER-, ER-/PR-, and premenopausal breast cancer, as well as having preventive effects on breast cancer in alcohol-consuming individuals [137].

Apart from glutamine, an increased serine/glycine metabolism, which is strongly linked to folate metabolism, is associated with increased tumour cell proliferation and an adverse outcome for patients [135]. Cancers typically have dysregulated immunity and tolerance, which can be manipulated by tryptophan and arginine [138]. Breast tumour settings increase arginase activity, the enzyme responsible for catalysing L-arginine, thereby affecting T cell adaptability, according to Z. Cavdar et al. [139].

### 6.3. Lipid Metabolism

According to C. Blücher et al. [140], fatty acids (FAs) and lipid metabolic programming are important factors that contribute to the development and spread of breast cancer. According to S. Beloribi-Djefaflia et al. [141], cancer cells exhibit active lipid and cholesterol metabolisms by promoting the absorption of exogenous lipids and lipoproteins or by bolstering de novo lipid and cholesterol manufacturing. Furthermore, de novo fatty acid synthesis (FAS) is primarily responsible for tumour cells’ heightened need for membrane metabolism to support their fast growth and multiplication. Fatty acid synthase (FASN), a crucial enzyme required for FAS, is expressed at higher levels in breast cancer [142]. Its upregulation is also linked to cancer development, recurrence, and a poor prognosis [143], indicating that increased FAS activity plays a role in the progression of breast cancer. Notably, FASN was reported to be the greatest in HER2-positive breast cancer and the lowest in TNBC at both the cell and tissue levels [144]. The “HER2–FASN axis”, a two-way regulatory mechanism between HER2 and FASN, has been postulated to improve the metastasis, proliferation, and chemoresistance of breast cancer [145]. By interacting with the FASN promoter region, the lipogenic transcription factor sterol regulatory element-binding protein (SREBP)-1 can control the expression of FASN [146]. FASN expression is likely regulated through the PI3K/AKT/mTOR and MAK signal transduction pathways, as suggested by [147]. The activation of AKT and SREBP-1 in breast tumour cells causes the FASN gene to be increased under hypoxic circumstances [148]. FASN expression in breast cancer cells can be reduced by both the mTOR inhibitor rapamycin and MAPK pathway inhibition [149].

## 7. Prognostic Prediction of Breast Cancer Facilitated by Metabolomics

Despite having enough oxygen, malignant tumour cells continue to engage in the glycolysis process. The “Warburg effect” refers to the observation that cancer cells prefer aerobic glycolysis over the more effective oxidative phosphorylation route [150]. According to U.E. Martinez-Outschoorn et al., high levels of lactates and ketones can hasten the growth and spread of cancer and may be useful indicators for predicting worse clinical outcomes in BC patients [151]. The most noticeable difference in the glucose metabolites between the malignant (BPLER) and less malignant (HMLER) TNBC cell lines was discovered to be Neu5Ac (N-acetylneuraminic acid). Subsequent investigation revealed that CMAS overexpression in HMLER enhanced the cells’ capacity to invade, whereas CMAS knockdown in BPLER significantly reduced the cells’ capacity to invade [152].

BC recurrence is associated with an increased risk of obesity [153]. Distant metastasis is reportedly initiated by 27-hydroxycholesterol, an oxysterol metabolite of cholesterol. The most decreased metabolites in highly invasive cell lines include sphingomyelin, PE, fatty acids, and a few unknown lipids. Membrane phospholipids, such as phosphatidylglycerol, phosphatidic acid, and phosphatidylethanolamine, are the most elevated [154]. Low levels of SM may be linked to reduced ceramide, which may have an impact on the apoptotic process, while high levels of PG, which may alter invasive ability, are thought to be correlated with mitochondrial malfunction [155]. The MCF-7, MCF-10A, and MDA-MB-231 BC cell lines were subjected to metabolic and lipidomic profiling. The results indicated the presence of a panel consisting of glucose-6-phosphate, xanthine, mannose6-phosphate, adenine, and guanine, which could potentially serve as a prognostic marker for BC metastasis. Lipidomic profiling showed higher phospholipid levels in metastatic groups compared to normal cells. Preliminary investigation revealed different expressions of 15 metabolites related to the phospholipid metabolism, fatty acid β-oxidation, and sphingolipid metabolism pathways in blood samples from 20 invasive ductal carcinoma patients and healthy controls [156].

The liver, brain, lungs, and bones are among the typical places where BC metastasises [157,158]. A prior team used LC-MS/MS-based targeted metabolomics and lipidomics to analyse plasma and track the progression of pulmonary metastasis in mice after inoculation with 4T1 metastatic BC cells. They discovered that changes in the arginase (ARG) pathway may indicate early-stage cancer metastasis and may be a target for future therapy [159]. According to K. Kus et al., modifications in glycolysis, lipid signalling, and structure may serve as potential biomarkers for aggressive cancer and late-stage metastasis. When examining the urine metabolites of BALB/c mice using NMR spectroscopy, a study team discovered that taurine, trimethylamine, creatine + phosphocreatine, and trimethylamine-N-oxide were all more plentiful in the early phase and that creatinine and allantoin had dramatically reduced [160]. Furthermore, during bone metastasis, there is a decrease in anti-metastatic lysophosphatidylcholines and an increase in pro-metastatic arachidonic acid [161].

Metabolomics can be used to measure survival results. According to N. Auslander et al., metabolites such as glucose, glycine, serine, and acetate have a substantial correlation with patient survival. Thus, metabolomics may be used to immediately assess the results of patients’ therapeutic effects [162].

## 8. Metabolomics Acts as a Diagnostic Detector of Breast Cancer

Even in cases where the deviation was mild, an alteration in metabolites might be identified at an early stage. Table 3 highlights the related studies undertaken in the respective experimental set ups. Transcriptional, translational, and molecular interactions can amplify even a slight abnormality in the body; metabolites are the end products of metabolic events that occur throughout the body. In [163], metabolomics was utilised as an early indicator of BC, using a dried-blood-spot-based direct infusion MS metabolomic analysis for rapid diagnosis. A diagnostic panel was created when 21 BC-related metabolites (asparagine, piperamide, proline, tetradecenoylcarnitine/palmitoylcarnitine, phenylalanine/tyrosine, and glycine/alanine) showed distinct changes in the blood. According to S. Wang et al. [164], this panel tested another set of BC and non-BC samples, demonstrating a sensitivity of 92.2% and a specificity of 84.4%. Additionally, there was a change in the blood’s amount of amino acid metabolites [165]. Furthermore, certain amino acids are found at reduced levels in the plasma samples of primary BC patients [166]. Significant variations were also seen in the blood amino acid and organic acid profiles of controls, benign patients, and BC patients. The most significant discovery is the difference in the taurine and glutamic acid levels between BC patients and controls—the taurine and glutamic acid levels are higher in the former and lower in the latter [167]. The tryptophan levels in blood samples from malignant tumours were noticeably greater than those from benign tumours [168]. Tyrosine, tryptophan, and creatine may act as a filter for both control and benign invasive ductal carcinoma (IDC) participants. A study found a significant decline in 9-cisRA from normal controls to metastatic BC. Oestrogen has two effects on BC, with oestrogen-induced BC models having higher levels of phosphatidylcholines and lysophosphatidylcholines. An ultra-high performance liquid chromatography-time-of-flight tandem mass spectrometer (QTOF-MS/MS) was used to compare serum metabolite profiles [169,170]. Therefore, the metabolite patterns in the blood may offer a useful way to distinguish BC patients from presumed healthy persons [171].

Another non-invasive sample is saliva. Salivary metabolite identification and analysis unique to British Columbia revealed variations in salivary metabolites related to cancer, although further research is still required [187]. Saliva samples from BC patients and healthy controls were analysed, and it was found that the following three additional groups were upregulated: lysophosphatidylcholine (18:1), lysophosphatidylcholine (22:6), and monoacylglycerol (0:0/14:0/0:0), and their AUC values were 0.920, 0.920, and 0.929, respectively [188]. Takayama et al. developed a diagnostic technique for BC patients based on saliva polyamine ratios and acetylated forms. The study found a high correlation between cancer patients and eight polyamines or their acetylated versions [189]. Dimethylheptanoylcarnitine and succinic acid were found to be considerably altered in the urine samples of healthy controls and BC patients among Hispanic women, according to an analysis conducted using LC-MS or GC-MS. Next, using a receiver operating characteristic analysis, the combination of these two metabolites was assessed with a 93% sensitivity and 86% specificity [190].

Nam et al. identified nine metabolic pathways based on the gene expression profiles associated with breast cancer by combining tissue transcriptomics and urine metabolomics. Ultimately, it was discovered that there were differences between normal and aberrant samples in the following four metabolic biomarkers: urea, 4-hydroxyphenylacetate, 5-hydroxyindoleacetate, and homovanillate [191]. These investigations showed that the metabolite measurement in urine was a potentially useful biomarker, at least for early BC detection. A prior team presented an untargeted metabolomics study using ultra-performance Liquid Chromatography in conjunction with a quadrupole time-of-flight (UPLC-QTOF) mass spectrometer to analyse the breast ductal fluid from both afflicted breasts and unaffected unilateral breast cancer. According to M. Phillip et al., breath volatile organic molecules may be a potential early detection method for breast cancer. These compounds, including 2-pentanone, 2-heptanone, and 3-methyl-3-buten-1-ol, can be identified in BC cell lines and normal human mammary epithelial cells. TNBC shares histological features with germline BRCA1-associated tumours, with lower levels of adenine, N6-methyladenosine, and 1-methyladenosine in patients with BRCA1 mutations [192]. Consequently, according to B. Roig et al. [193], these metabolites might be prospective BC biomarkers connected to BRCA1 mutations [194,195].

## 9. Molecular Hallmarks of Triple-Negative Breast Cancer in Relation to Metabolic Reprogramming

To determine the processes that propel carcinogenesis, it is crucial to characterise the molecular characteristics of TNBC. It is possible to achieve these carcinogenic effects by interfering with TNBC cells’ metabolic reprogramming. One important molecular process in TNBC cells is MYC amplification [196]. The paper also reports that 40% of TNBC overexpresses the MYC oncogene, which encodes c-myc, a transcription factor linked to metabolic reprogramming and carcinogenesis. To bind certain DNA sequences and control gene expression for its transcriptional activity, C-myc interacts with MYC-associated factor X (Max), another helix–loop–helix leucine zipper protein [197,198].

It was found that C-myc can transactivate metabolic pathways genes and collaborate with other drivers like HIF-1α, supporting cellular survival. MYC amplification is, therefore, essential for TNBC cells’ metabolic reprogramming to promote carcinogenesis. With a frequency of up to 80%, p53 is the gene most often altered in TNBC [195]. In response to metabolic stress signals, the tumour suppressor p53 is essential for preserving genomic integrity [198]. Nonetheless, p53 mutations result in a protein with a reduced capacity to bind to certain DNA sequences, resulting in a dysregulated p53 transcriptional pathway [199]. AMP-activated protein kinase (AMPK) phosphorylation activates p53. When p53 is activated in nutrient-deprived environments, it triggers a reversible cell-cycle checkpoint response [200]. On the other hand, cancer cells with p53 mutations that lack this response would be better able to proliferate when challenged with a shortage of nutrients. Another significant molecular change that has recently been identified in TNBC cells is the deletion of Beclin-1 [201]. Beclin-1 (BECN1) is a tumour suppressor that has been shown to increase autophagy by participating in the lysosomal degradation pathway. It is expressed at a lower level in TNBC and other breast carcinomas [202]. Scavenging unfolded proteins and damaged organelles, such as mitochondria, using autophagy-related proteins is a crucial aspect of autophagy. Furthermore, by the modification of important metabolic enzymes, autophagy-related proteins can control metabolic reprogramming. For instance, decreased autophagic activity increases glycolysis, which helps cancer cells to survive [203]. Therefore, BECN1-related autophagy may be a negative regulator of metabolic rewiring that promotes the development of breast cancer. Relentlessly, autophagy and BECN1 depletion might be the connection between metabolic reprogramming and carcinogenesis in TNBC. TNBC differs from other subtypes of breast cancer in that it possesses molecular characteristics such as a greater loss of PTEN, a lower PIK3CA mutation, and RB1 expression. These features are important because they give TNBC cells a unique metabolic phenotype that helps them to survive under metabolic stress [195].

## 10. Metabolism-Targeting Drugs in Metastatic Breast Cancer

Patients with breast cancer that have not yet discovered metastases are highly likely to do so, and because there are currently no viable therapies for metastatic breast cancer, these tumours cannot be cured. A patient’s chances of survival will be increased by early intervention during the colonisation and growth phases of distant metastases. Numerous intriguing drugs that target altered metabolic pathways are being investigated in various preclinical and clinical phases [204]. Nevertheless, the therapeutic utility of metabolic-interfering medications in the management of BC remains unclear at present. Adding indoximod, an inhibitor of the Indoleamine 2,3-dioxygenase 1 (IDO1) pathway, to taxane did not increase PFS in patients with HER2-metastasised BC as compared to taxane alone, according to a Phase II clinical study comprising 164 patients [205]. To counteract chemotherapy resistance, the use of glucose metabolism inhibitors such as metformin and 2-deoxy-D-glucose (2-DG) in conjunction with chemotherapy has shown positive results [206]. A verified partial response (PR) lasting 65 days was observed in one patient with medullary breast cancer that had spread to the lymph nodes and lung who was receiving therapy with 45 mg/kg 2-DG every other week [207]. Dichloroacetate (DCA) can reduce metformin-induced lactate production through PDK1 inhibition while also increasing metformin-induced oxidative damage. This suggests that novel combinations, like DCA and metformin, will be promising in expanding the therapies for breast cancer [208]. Nevertheless, additional investigation is required because of the small sample size and the paucity of beneficial data.

The combination of metabolic inhibitors and checkpoint inhibitors has the potential to improve the effectiveness of immunotherapy. Studies are now being conducted to determine the connection between tumour-intrinsic metabolism and effective immunotherapy. T cell hypo responsiveness during cancer may be mediated by the metabolic constraints imposed by tumours. Checkpoint blockade antibodies against PD-1, CTLA-4, and PD-L1 can replenish the glucose in the tumour microenvironment, allowing T cells to produce interferon and glycolysis. Conversely, directly blocking PD-L1 on tumours reduces glycolysis by mTOR inhibition and glycolysis enzyme expression reduction [209]. Given the growing popularity of immunotherapy for breast cancer, future studies must focus on the metabolic relationship between cancer and immune cell infiltration.

Endocrine therapy is particularly useful for individuals with ER-positive breast cancer; nevertheless, some patients will develop resistance to endocrine therapy. According to Y. Zhao et al., there is evidence that trastuzumab-resistant HER2+ breast malignancies have an elevated glycolysis phenotype, and that glycolytic restriction can make these tumours more susceptible to trastuzumab therapy [210]. Since there are no therapeutic targets for TN/basal-like breast cancer, chemotherapy is presently the primary course of treatment. Many of the current studies concentrate on metabolic interference in metastatic preclinical models and chemotherapy resistance models because of the distinct metabolic profile of TNBC [211]. Since the metabolic properties of tumour cells and their surroundings vary throughout metastatic locations, therapeutic strategies that target such differences may also be considered in the future [212]. According to Y. Zhao et al., there is evidence that trastuzumab-resistant HER2+ breast malignancies have an elevated glycolysis phenotype, and that glycolytic restriction can make these tumours more susceptible to trastuzumab therapy [210].

## 11. Metabolomics and Treatment Implications for Breast Cancer

In terms of treatment and therapeutic interventions for BC using metabolome research, these studies are still in progress, and there is an imperative urge to translate these bench studies to the bedside. The advent of high-throughput metabolomics techniques, like lipidomics and global untargeted metabolomics, has made it possible to measure metabolite abundance directly [213]. The following three categories of metabolic targets exist for cancer treatment: body metabolism, tumour cell metabolism, and tumour microenvironment metabolism. Metabolomics will provide a deeper understanding of the pathways and acting metabolites in breast cancer, which could further help in the development of precision medicine [214,215].

Based on the important and basic principles of molecular biology, OMICs is a neologism majorly used in biomedical research, which includes datasets from genomics, transcriptomics, proteomics, and metabolomics [216]. Metabolomics is one of the OMICs sciences and is a useful tool that offers cutting-edge analytical instrumentation in combination with pattern recognition techniques and chemometric tools to search for new disease biomarkers that offer fresh perspectives on the aetiology of diseases and more reliable evaluations of aetiological pathways [217].

For instance, rewiring the T cell metabolism has great promise in cancer therapy, as T cells are essential for antitumour immunity. It was determined by earlier research that glycolysis was crucial for T cell activation [218]. Interestingly, it has been demonstrated that arginine causes activated T cells to switch from glycolysis to OXPHOS, which enhances the development of central memory-like cells that have a greater ability to survive and exhibit potent antitumour activity in vivo. Furthermore, it was shown that monocarboxylate transporter 1 (MCT1) is a possible target for immunotherapy, since it allows regulatory T cells (Tregs) to absorb lactate and increase the expression of PD-1 [219]. It has recently been demonstrated that blocking K+ channels and aryl hydrocarbon receptors causes TAMs to switch from OXPHOS to glycolysis, which makes cancers more susceptible to immunotherapy [220].

In one study, TNBCs were categorised into the following three distinct metabolomic subgroups: C1, which was found to have an enrichment of ceramides and fatty acids; C2, which showed an upregulation of the metabolites associated with glycosyl transfer and oxidation reaction; and C3, which had the least amount of metabolic dysregulation [42]. The TNBC transcriptomic subtypes were refined and identified. Research using xenograft and organoid models derived from patients suggests that sphingosine-1-phosphate (S1P), an intermediate in the ceramide pathway, is a potential target for LAR tumour treatment. The study also concluded that N-acetyl-aspartyl-glutamate is an important metabolite that promotes tumour growth and is a possible target for treatment in high-risk BLIS tumours.

Another study by E. Pietri et al. focused on androgen receptors in patients with triple-negative breast cancer. They examined the effects of oral androgen precursor dehydroepiandrosterone (DHEA) intake at a dose of 100 mg/day on post-menopausal patients with breast cancer. There was a control group and an intervention during the eight-week course of treatment [219].

As metabolites have a direct influence on lifestyle, diet, environment, etc., one study also noted the effects of a multi-nutrient supplement on breast cancer patients who were scheduled for surgery following a biopsy-confirmed diagnosis. Patients with breast cancer took three capsules of natural extract every day. The capsules with multiple nutrients included extracts of pomegranates, oranges, lemons, olives, cocoa, and grape seeds [221]. As a result, changes in 2urolithin A-3-O-glucuronide, 2,5-dihydroxybenzoic acid, and resveratrol-3-O-sulfate, which is an oncometabolite, were observed using UPLC-ESI-QTOF-MS. Another intervention in diet included, based on the CRD, the fasting mimicking diet (FMD), with alternating weeks of extremely low-calorie intake for five days followed by a period of no calorie restriction for the rest of the month [14].

To prevent the onset and spread of cancer, the FMD lowered the levels of circulating insulin-like growth factor 1, leptin, and insulin. Furthermore, there has been increased interest in the ketogenic diet (KD) as a possible cancer treatment in recent years [221]. The KD is a high-fat, adequate-protein, and very-low-carbohydrate regimen that induces the production of ketone bodies and mimics the fasting state. In the study by [222], intravenous selenium injections that affected patients with breast cancer’s metabolism were studies. It was concluded that the experimental group had significantly higher levels of the anti-inflammatory chemicals corticosterone, LTB4-DMA, and PGE3 than the control group did by employing the use of UHPLC-Q-Exactive Orbitrap/MS.

## 12. Clinical Implications of Metabolomics for Breast Cancer

Metabolomic-guided diagnostics for BC studies have been thoroughly experimented with in cell lines and animal models, leading to preclinical studies, but very few studies have made their way to clinical trials. For instance, the study of metformin targeting the OXPHOS pathway with the ETS complex as a target made its way to Phase III of the clinical trial, but could not make it through due to a lack of significant results [34].

Still, there are challenges related to inducing the bench-to-bedside transition for such an interdisciplinary field, which could be subjected to high-cost experiments and failure in relevant fieldwork due to the unpredictable behaviour of metabolites. In [43], the challenges and limitations associated with metabolomics and their use in clinics were highlighted. Many obstacles must yet be removed before metabolomics can be widely used in clinical research and practice, even though both untargeted and targeted metabolomics have enormous potential in biomarker discovery and hypothesis testing in the translational setting. As previously mentioned, in order to cover the entire metabolism, multiple complementary methods are needed. Many times, this calls for several instrumentation platforms, which many clinical and academic labs might not have. Additionally, a variety of software programs are available for processing and analysing data, especially in untargeted metabolomics. A robust experimental design is also necessary for analysing large metabolomics data sets, as this enables suitable statistical analysis. Thus, to conduct a successful metabolomics research study, at the very least, analytical chemists, statisticians, and biologists are needed. Metabolomic investigations have the potential to expand classifications and provide extra therapeutic benefits by confirming distinctions across breast cancer subgroups based on genetic, histological, or clinical data. Data from a combination of HR-MAS MRS and gene expression microarrays conducted on 46 individuals with early-stage breast cancer provide unambiguous support for this idea [44]. Using metabolomic analysis, luminal A breast cancers were subtyped into three groups based on differences in α-glucose, β-glucose, amino acids, myo-inositol, and lipid residues. Specifically, tumours with elevated glycolytic activity were included in a category known as A2, which was defined by lower glucose levels and greater alanine levels. This kind of research provides credence to the idea that the metabolomic profiling of breast cancers may be an extra layer worth investigating in our quest for more individualised treatment strategies.

## 13. Public Health Implications of Healthcare Policies, Screening Programs, Improved Patient Care, and Quality of Life for Individuals with Breast Cancer

Research, education, and the promotion of the highest quality of patient care are important to create a world where cancer is prevented and every survivor is healthy. The community–clinical linkage is one of the most important areas to focus on when implementing screening activities, as today, the CCL plays an important role in the prevention and management of chronic diseases such as cancers, cardiovascular diseases, and even high-blood-pressure-related issues. In its policy statement on cancer inequalities in the year 2009, ASCO addressed the variations in cancer outcomes. Cancer disparities, which mean measurable differences in cancer outcomes in various population groups, have persisted throughout the past few years, despite the advancements in detection and treatment methods and protocols, as well as modifications in systems that provide cancer care, such as the enactment of the Affordable Care Act. Factors like race/ethnicity, geography, gender identity, and sexual orientation have played roles in broadening our understanding of the communities facing disparate results [51]. All populations that stand to gain must be represented in research used to produce new advancements in cancer care, which has been growing and becoming increasingly intricate and individualised. Gains in attaining health equity will be constrained if new developments are created through research that excludes representation from all populations, since many populations lack access to fundamental, evidence-based care [223]. To achieve this, stakeholders—including patients, caregivers, clinicians, policymakers, pharmaceutical companies, and advocacy groups—can collaborate to create suitable, focused strategies. It is more likely that research will recognise and possibly alleviate health inequities if demographic and clinical data are routinely collected and reported [224]. To drive equitable inclusion in research, healthcare professionals and stakeholders should engage in meaningful and ongoing partnerships with private and public entities composed of healthcare professionals. It is important to drive the equitable inclusion in research of academic and community practices, patients, caregivers, and other organisations. Despite a slight improvement in the understanding of cancer health disparities over the last 10 years, educational initiatives should focus on the research, policies, programs, and activities that have been effective in reducing these disparities. Multisector organisations and stakeholders must guarantee cultural literacy awareness and supply relevant literacy materials as part of public awareness and information campaigns [225].

## 14. Challenges

Despite its potential, metabolomics still has several inherent drawbacks that have significantly hindered its broad application in clinical settings thus far. Interindividual variances among patients, sample variability, and a significant lack of established techniques for tissue processing are among the major constraints that stem from both biological and experimental variables. Furthermore, biological elements like warm and cold ischemia could significantly affect the outcomes of omics-based research. It is difficult to forecast the consequences of a tissue specimen’s exposure to warm ischemia brought on by artery ligation and body excision, since these outcomes rely on the kind of illness and the surgical technique used [49]. However, to guarantee high-quality omics data, the impacts of various cold ischemia periods (i.e., the intervals from resection to fixation and/or to freezing for cryo-preservation) were examined. According to recent research, samples of breast tumours frozen within 30 min following excision showed no appreciable changes in the composition of individual metabolites. Following this period, there were some metabolic alterations in the levels of ascorbate, creatine, and glutathione as well as in the content of phospholipid metabolites [226]. Our group’s earlier work showed that tissue specimens for histological, transcriptomic, and proteomic analyses can be stored for up to 48 h under vacuum storage (UVS), but this approach has drawbacks for metabolomic applications, since we discovered that, in contrast to other omics techniques, the metabolome is more impacted by the storage duration, as we observed an increase in the content of free choline in both normal and malignant breast tissue during vacuum storage.

## 15. Conclusions and Future Prospects

Cancer elimination in the twenty-first century will most likely rely not only on more effective customised therapy, but also on earlier identification and prevention. As the most frequent disease in women globally, breast cancer has a significant impact on contemporary society, making it imperative that new technologies be developed to better comprehend the molecular alterations involved. The current paucity of trustworthy biomarkers indicating metabolic changes emphasises the necessity of identifying early prognostic and diagnostic markers for breast cancer. Metabolomics offers the potential to provide new, low-cost, non-invasive diagnostic procedures based on disease biomarkers that are straightforward and have a good sensitivity and specificity. To enable the early detection of breast cancer, these putative indicators should be further verified in subsequent research using a sizable patient cohort. Finding the pathways that are critical for tumour growth and treatment response, as well as the biomarkers that may be used to track these pathways’ activity, is crucial for improving patient stratification. Improved comprehension of worldwide disruptions in metabolic processes may yield a significant understanding of disease aetiology, prognostic factors, and diagnostic biomarkers. Through enhanced molecular diagnostics that result in better therapeutic approaches through an assortment of effective drugs as part of systems medicine, new insights into the mechanisms behind the progression of cancer and the treatment of cancer patients will be provided. This increases the possibility of finding and validating true metabolic biomarkers for breast cancer. Eventually, this will result in diagnostic toolkits that enable far more accurate prognostic and predictive analyses.

## Figures and Tables

**Figure 1 cells-14-00005-f001:**
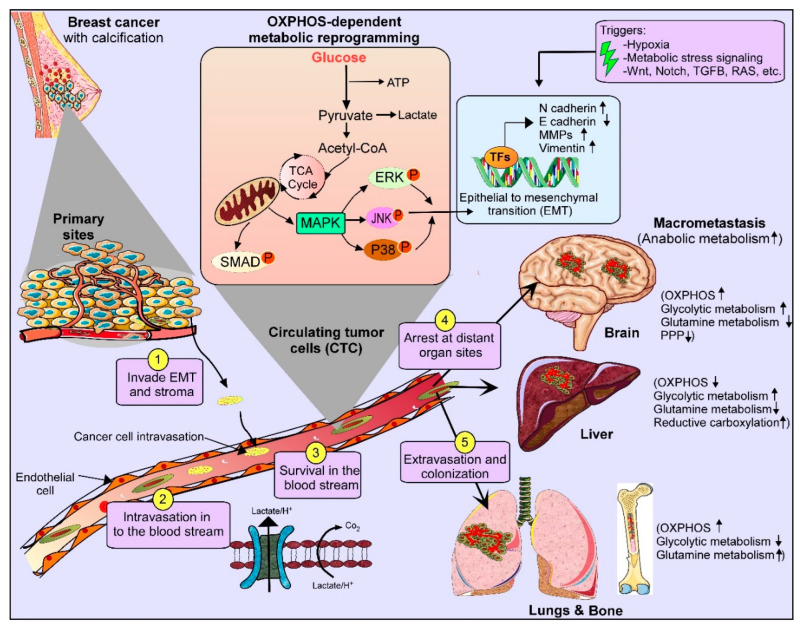
A summary of potential non-invasive biomarkers for the early detection of breast cancer using fluid samples as the source. Abbreviations: MAPK: Mitogen-Activated Protein Kinase, ERK: Extracellular Signal-Regulated Kinase, JNK: c-Jun N-terminal Kinase, EMT: Epithelial–Mesenchymal Transition, OXPHOS: Oxidative Phosphorylation.

**Figure 2 cells-14-00005-f002:**
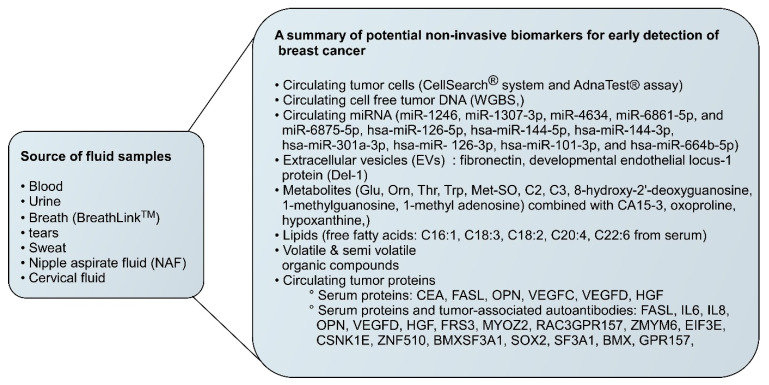
Breast cancer metastasis involves multiple steps, starting with the invasion of surrounding tissues, followed by intravasation into the bloodstream or lymphatics, circulation through the body, and extravasation into distant tissues. Various factors, such as EMT, MET amplification, and abnormal TGFb production, can contribute to tumour metastasis. Abbreviations: CEA: Carcinoembryonic Antigen, FASL: Fas Ligand, OPN: Osteopontin, VEGFC: Vascular Endothelial Growth Factor C, VEGFD: Vascular Endothelial Growth Factor D, HGF: Hepatocyte Growth Factor, FRS3: Focal Adhesion Kinase Related-3, MYOZ2: Myozenin 2, RAC3GPR157: Rac Family Small GTPase 3 G Protein-Coupled Receptor 157, ZMYM6: Zinc Finger MYM-Type Protein 6, EIF3E: Eukaryotic Translation Initiation Factor 3 Subunit E, CSNK1E: Casein Kinase 1 Epsilon, ZNF510: Zinc Finger Protein 510.

**Figure 3 cells-14-00005-f003:**
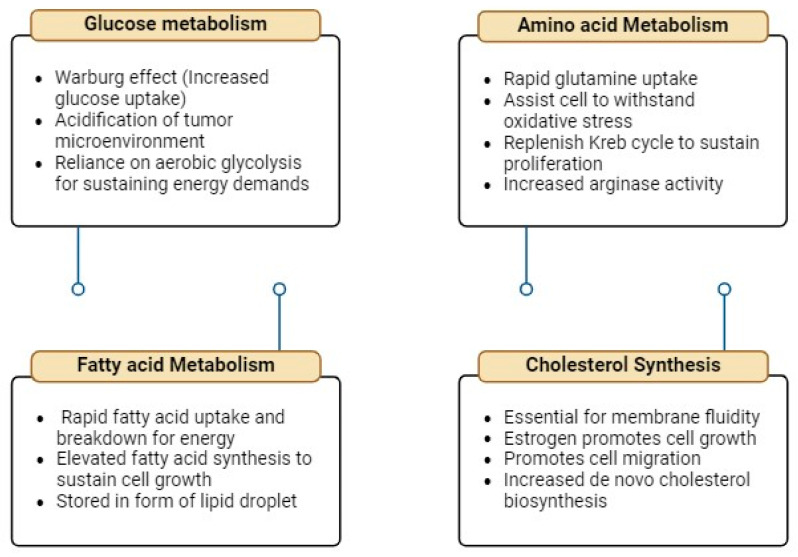
Schematic representation of various metabolic pathways involved in breast cancer prognosis.

**Table 1 cells-14-00005-t001:** Investigation of breast cancer through omics-based technologies [34].

Central Dogma of Cancer Biology	Omes	Omics	Technologies
ctDNA, mtDNA, nDNA	Genome	Genomics	DNA sequencing, Whole genome sequencing, DNA microarray, targeted gene sequencing
DNA methylation	Methylome	Epigenomics	Microfluidics assays, methylation analysis sequencing, ChIP sequencing
mRNA, ncRNA, miRNA, lncRNA	Transcriptome	Transcriptomics, miRomics	qRT-PCR, whole transcriptome analysis, RNA microarray
mRNA, rRNA, tRNA	Translatome	Translatomics	Polysome profiling, 2-dimensional electrophoresis, HPLC, FRET
Peptide, protein	Proteome, phosphoproteome, glycoproteome, interactome	Proteomics, phosphoproteomics, glycoproteomics, interactomics	LC-MS/MS, LC-ESI-MS/MS, MALDI-ToF MS
Metabolites, lipids	Metabolome, lipidome	Metabolomics, lipidomics	NMR, LC-MS, GC-MS, DESI-MSI

Abbreviations: ctDNA: circulating tumour DNA, mtDNA: mitochondrial DNA, nDNA: nuclear DNA, mRNA: messenger RNA, ncRNA: non coding RNA, miRNA: micro RNA, lncRNA: long non-coding RNA, rRNA: ribosomal RNA, tRNA: transfer RNA, HPLC: High-Performance Liquid Chromatography, FRET: fluorescence resonance energy transfer, NMR: Nuclear Magnetic Resonance, LC-MS: Liquid Chromatography–Mass Spectrometry, GC-MS: Gas Chromatography–Mass Spectrometry, DESI-MSI: Desorption Electrospray Ionization–Mass Spectrometry Imaging, LC-ESI-MS/MS: Liquid Chromatography–Electrospray Ionization–Mass Spectrometry, MALDI-ToF MS: Matrix-Assisted Laser Desorption Ionization–Time of Flight Mass Spectrometry.

**Table 2 cells-14-00005-t002:** Omics-based investigations of BC samples using different technologies (adapted from [42]).

Sample Type	Technique Used	Changes in Metabolites	Reference
Breast cancer tissue	GC-MS LC-MS	The metabolites involved in glycolysis, glycogenolysis, TCA cycle, proliferation, and redox pathways, such as the NAD+ synthesis pathway, were found to be higher in TNBC in comparison to ER + ve.	[43]
Breast cancer tissue	UPLC-MS/MS	Breast cancer tissue samples had higher levels of membrane phospholipids (phosphatidylcholine, phosphatidylethanolamine, and sphingomyelin ceramides) than normal breast tissue (more so in ER-ve samples).	[44]
Breast cancer tissue	GC-MS LC-MS	Glutathione pathway intermediaries, tryptophan metabolite, 2-hydroxyglutrate onco-metabolites, glycolytic and glycogenolytic intermediaries, and the level of kynurenine were higher in ER-ve tumours than ER + ve ones.	[45]
Serum sample from breast cancer patient	LC-MS	Obese patients with breast cancer serum samples had much higher levels of lipid, carbohydrate, and amino acid metabolites; oxidative phosphorylation; uric acid; ammonia recycling; and vitamin metabolism (all of which play a role in ATP generation). Serum levels of neurotransmitter metabolites, including acetylcholine, histamine, and serotonin, were higher in obese breast cancer patients than in non-obese patients.	[46]
Plasma from breast cancer patient	LC-MS	In comparison to healthy controls, the plasma of breast cancer patients had higher levels of antioxidative metabolites (taurine and uric acid), bioenergetic metabolites (fatty acids capric acid, and myristic acid), three branched-chain amino acids that supply carbon for gluconeogenesis (2-hydroxy-3-methylbutiric acid, 2-hydroxy-3-methylpentanoic acid, and 3-methylglutaric acid), and substrates for nucleic acid biosynthesis (cystidine and inosine diphosphate).	[47]
Plasma from breast cancer patient	LC-MS	In the plasma of breast cancer patients, arginine proline metabolism pathway metabolites, tryptophan metabolism pathway metabolites, and fatty acid biosynthesis pathway metabolites were higher than in normal healthy individuals.	[48]
Breast cancer tissue	HR MAS MRS	A2 had a lower glucose signal than A1 and A3. In comparison to A2, the _-hydrogen amino acid signal was higher in A3 and lower in A1. The signal for alaanine was greater in A2 than in A3. A1 had a lower myo-inositol signal than A2 and A3.	[49]
Serum sample from breast cancer patient	NMR LC-MS	Four metabolites were found, and the levels of glutamine and tryptophan dropped in the pCR group relative to the SD group. In comparison to SD and PR, the pCR group had higher levels of isoleucine and lower levels of linolenic acid.	[50]
Fasting blood (serum and plasma) sample from healthy and breast cancer patients	LC-TOF-MS GC-TOFMS	The metabolite for the taurine pathway (pyruvate and hypotaurine), which is also the metabolite for glycine, was increased in breast cancer compared to normal healthy individuals, while the levels of phospholipid biosynthesis metabolite, glycerol 3 phosphate, and the amino acids succinate, choline, serine, glycine, and alanine are lower in breast cancer patients’ serum and plasma samples than in those of healthy individuals in normal circumstances.	[51]

Abbreviations: LC-MS: Liquid Chromatography–Mass Spectrometry, GC-MS: Gas Chromatography–Mass Spectrometry, UPLC-MS/MS: Ultra-Performance Liquid Chromatography–Tandem Mass Spectrometry, HR MAS MRS: High-Resolution Magic Angle Spinning Magnetic Resonance Spectroscopy, LC-TOF-MS: Liquid Chromatography–Time of Flight-Mass Spectrometry, GC-TOFMS: Gas Chromatography–Time of Flight Mass Spectrometry, NMR: Nuclear Magnetic Resonance.

**Table 3 cells-14-00005-t003:** Summarisation of experiments conducted in vitro/in vivo models employing the use of metabolomics.

Type of Experiment	Model Used	Technique Employed for Detection	Sample Type	Observations from the Study	Reference
In vitro	MCF-7 and MCF-10A	LC-TOFMS and GC-TOFMS	Cell lysate	Aspartate level was low	[172]
In vitro	MCF-7 and MDA-MB-231	NMR	Cell lystate	Decreased glucose uptake in both cell lines was found to be an effect of metabolic disorders caused by inositol 1,4,5-trisphosphate receptors (IP3R)	[173]
In vivo	MCF-7, MCF-7/D40, MDA-MB-231 cells in SCID mice	P-MRS, H NMR	Serum	Choline compounds were found to be altered	[174]
In vivo	MDA-mb-468 cells in CD-1 Nude mice	HPLC/MS	Tumour/plasma	Both tumour and KD impact AA metabolism and FA transportation, with KD reversing the metabolic signature of BC mice	[175]
In vivo/ex vivo	MDA-MB-231, MDA-MB-231-GDPD5-shRNA cells in athymic Nude mice	NMR	Serum	GDPD5 silencing was found to be upregulated in the in vivo study and PE GDPD5 was found to be upregulated in the ex vivo study	[176]
In vivo	MCF-7 cells in athymic BALB/c Nude mice	HRMS	Tumour tissue	Creative, GPC, and PC levels were found to decrease	[177]
In vivo	NMuMG–NT2197 cells in Nude mice	GC/MS	Tumour extract	An increase was observed in lactate, fumarate, malate, α-KG, citrate; Phenformin/Lapatinib	[178]
In vivo	MCF-7 cells in athymic BALB/c Nude mice	GC/GC-TOF/MS; UPLC-QTOF/MS	Urine/serum	Increased anabolism corelated in urine samples and increases in fumarate, 2-OG, and succinate in serum samples	[179]
In vivo	MDA-MB-231 cells in NIH-III Nude mice	HNMR	Serum	Increased levels of leu, lys, phe, thr, 1-methyl-his, 2-HB, lactate, and pyr and decreased 2-OG	[180]
Patient-derived xenograft model	Orthotopic implantation of MAS98.12 and MAS98.06 in SCID mice	HRMAS	Tumour tissue	Increased levels of glycine in basal-like tumours and decreased level of glycine in luminal-like tumour	[181]
Patient-derived xenograft model	Orthotopic implantation of MAS98.12 and MAS98.06 in SCID mice	H HRMAS	Tumour tissue	Increased glyciene in untreated basal-like tumours. Decrease in lactate and BEZ235	[182]
In vitro	MCF-7	Immunoblot assay	Cell lysate	Sulphur amino acid metabolism was identified	[183]
In vitro	MCF-7	GC/MS	Cell lysate	Decrease in glutathione biosynthesis and increase in glycerol metabolism	[184]
In vitro	BT474 MCF-7 MDA-MB-231 MDA-MB-468	NMR	Cell lysate	Paclitaxel increased myo-inositol and decreased lactate and creatine in luminal A cell lines. In drug-treated TNBC cell lines, glutamine, glutamate, and glutathione increased and lysine, proline, and valine decreased	[185]
In vitro	MDA-MB-231	HR-MAS-NMR		Alterations in lactate phosphocholine and acetate	[186]

## Data Availability

Not applicable.

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
