# Peer review of "Metabolomics-Driven Biomarker Discovery for Breast Cancer Prognosis and Diagnosis"

_cells, 2024, doi:10.3390/cells14010005_

Round 1

Reviewer 1 Report

Comments and Suggestions for Authors

The current article under review has summarized the important literature revolving around -omics technology in relation to breast cancer biomarkers. "Cancer elimination in the twenty-first century will most likely rely not only on more effective customized therapy, but also on earlier identification and prevention", this statement is important as early intervention has been shown to be extremely effective, especially with the alarming rise in the rates of BC in young women. Overall, the article is well written and has highlighted some important developments in the field and the detailed focus on metabolomics work will be of interest to the journal's readership. I recommend addressing some the comments below. 

Major concerns:

1. The authors are requested to review the following articles for breast cancer biomarker in non-traditional biofluids. The current article has addressed saliva, urine, and breast milk. However, there are articles that are focused on using other biofluid sources such as tears and nipple aspirate fluids. Though some of these are highlighted in figure 2, the proper references have not been provided.
a) Daily, Anna, et al. "Using tears as a non-invasive source for early detection of breast cancer." PLoS One 17.4 (2022): e0267676.
b) 
Inubushi, Sachiko, et al. "Oncogenic miRNAs identified in tear exosomes from metastatic breast cancer patients." Anticancer research 40.6 (2020): 3091-3096.
c) Shaheed, Sadr-ul, et al. "Evaluation of nipple aspirate fluid as a diagnostic tool for early detection of breast cancer." Clinical proteomics 15 (2018): 1-15.

2) I assume that the authors wanted to highlight the section after Figure 1 as a separate section but might have missed out on adding a title for section 4. Currently, the article jumps from 3 to 5. It makes sense to break down section 3 into two different sections. Additionally, the authors are recommended to talk about more recent technological advancements in proteomics such as micro-arrays (somalogic which can perform up to 11,000 proteins with minimal samples), and proximity extension assays (Olink).

Minor concerns:

1. The authors are recommended to have a native English speaker review the article for the use of proper punctuations at multiple locations throughout the article.  

2. Instead of saying "According to [reference number], please state the group that conducted the study or alternatively, use "According to 'author name' et al." 

3. Lines 467-471, check for font. 

Author Response

General Comment:

“The current article under review has summarized the important literature revolving around -omics technology in relation to breast cancer biomarkers. "Cancer elimination in the twenty-first century will most likely rely not only on more effective customized therapy, but also on earlier identification and prevention", this statement is important as early intervention has been shown to be extremely effective, especially with the alarming rise in the rates of BC in young women. Overall, the article is well written and has highlighted some important developments in the field and the detailed focus on metabolomics work will be of interest to the journal's readership. I recommend addressing some the comments below.” 

Response:

Thank you very much for your valuable suggestions. We greatly appreciate the time and effort you have put into reviewing our manuscript. We have highlighted all the changes in yellow colour in the revised manuscript.

Comment 1:

“Major concerns:
1. The authors are requested to review the following articles for breast cancer biomarker in non-traditional biofluids. The current article has addressed saliva, urine, and breast milk. However, there are articles that are focused on using other biofluid sources such as tears and nipple aspirate fluids. Though some of these are highlighted in figure 2, the proper references have not been provided.
a) Daily, Anna, et al. "Using tears as a non-invasive source for early detection of breast cancer." PLoS One 17.4 (2022): e0267676.
b) Inubushi, Sachiko, et al. "Oncogenic miRNAs identified in tear exosomes from metastatic breast cancer patients." Anticancer research 40.6 (2020): 3091-3096.
c) Shaheed, Sadr-ul, et al. "Evaluation of nipple aspirate fluid as a diagnostic tool for early detection of breast cancer." Clinical proteomics 15 (2018): 1-15.”

Response:

We have added the new references as suggested by the reviewer.

Comment 2:

“2) I assume that the authors wanted to highlight the section after Figure 1 as a separate section but might have missed out on adding a title for section 4. Currently, the article jumps from 3 to 5. It makes sense to break down section 3 into two different sections. Additionally, the authors are recommended to talk about more recent technological advancements in proteomics such as micro-arrays (somalogic which can perform up to 11,000 proteins with minimal samples), and proximity extension assays (Olink).”

Response:

We have divided the sections and added information on technological advancements as suggested.

Comment 3:

“Minor concerns:

1. The authors are recommended to have a native English speaker review the article for the use of proper punctuations at multiple locations throughout the article.” 

Response:

The manuscript has been reviewed by native English speaker as suggested by the reviewer.

Comment 4:

“2. Instead of saying "According to [reference number], please state the group that conducted the study or alternatively, use "According to 'author name' et al.”

Response:

The references have been cited as per the suggestions.

Comment 5:

“3. Lines 467-471, check for font.” 

Response:

Font has been rectified as suggested.

Reviewer 2 Report

Comments and Suggestions for Authors

The article titled "Metabolomics-Driven Biomarker Discovery for Breast Cancer Prognosis and Diagnosis" by Rasanpreet Kaur et al. presents valuable insights; however, there are several areas that require attention:

  1. The authors primarily focus on the potential benefits and advancements in metabolomics for breast cancer research; however, it lacks a discussion on potential drawbacks or challenges of this approach.
  2. The authors can include the abbreviations used in the tables and figures under the respective tables or figures.
  3. Authors could provide more discussion on metabolomics and its treatment implications in various subtypes of breast cancer.
  4. Authors could create a comprehensive table based on in vitro and in vivo studies related to metabolomics as a diagnostic tool for breast cancer.
  5. Authors may include a list of metabolites used in breast cancer treatments based on clinical trials.
  6. Authors may expand on the public health implications of metabolomics studies in breast cancer, including healthcare policies, screening programs, improved patient care, quality of life for individuals, and personalized treatment.
  7. Authors can emphasize the significance of education and knowledge dissemination to foster high engagement and collaboration in advancing metabolomics studies in breast cancer.

Author Response

Comment 1:

            “The article titled "Metabolomics-Driven Biomarker Discovery for Breast Cancer             Prognosis and Diagnosis" by Rasanpreet Kaur et al. presents valuable insights;           however, there are several areas that require attention:

  1. The authors primarily focus on the potential benefits and advancements in metabolomics for breast cancer research; however, it lacks a discussion on potential drawbacks or challenges of this approach.”

Response:

The drawbacks of the metabolomics approach have been discussed in a separate section in the revised manuscript.

            Comment 2:

  1. “The authors can include the abbreviations used in the tables and figures under the respective tables or figures.”

Response:

We have included the abbreviations as suggested.

Comment 3:

  1. “Authors could provide more discussion on metabolomics and its treatment implications in various subtypes of breast cancer.”

Response:

Metabolomics and its treatment implications in various subtypes of breast cancer has been elaborated in the revised manuscript.

Comment 4:

  1. “Authors could create a comprehensive table based on in vitro and in vivo studies related to metabolomics as a diagnostic tool for breast cancer.”

Response:

Table 3 has been included to address the in vitro and in vivo studies related to metabolomics.

Comment 5:

  1. “Authors may include a list of metabolites used in breast cancer treatments based on clinical trials.”

Response:

As the reviewer suggested, we have the list of metabolites in our revised manuscript.

Comment 6:

  1. “Authors may expand on the public health implications of metabolomics studies in breast cancer, including healthcare policies, screening programs, improved patient care, quality of life for individuals, and personalized treatment.”

Response:

We have included the separate section on public health implications in the revised manuscript.

Comment 7:

  1. “Authors can emphasize the significance of education and knowledge dissemination to foster high engagement and collaboration in advancing metabolomics studies in breast cancer.”

Response:

We really thank the reviewer for this valuable suggestion. Taking the reviewer’s suggestion, we have included this in our revised manuscript.

Round 2

Reviewer 2 Report

Comments and Suggestions for Authors

Accept in present form